

# Novel rapid identification of Severe Acute Respiratory Syndrome Coronavirus 2 (SARS-CoV-2) by real-time RT-PCR using BD Max Open System in Taiwan

Cherng-Lih Perng[1,*], Ming-Jr Jian[1,*], Chih-Kai Chang[1],
Jung-Chung Lin[2], Kuo-Ming Yeh[2], Chien-Wen Chen[3],
Sheng-Kang Chiu[2], Hsing-Yi Chung[1], Yi-Hui Wang[1], Shu-Jung Liao[1],
Shih-Yi Li[1], Shan-Shan Hsieh[1], Shih-Hung Tsai[4], Feng-Yee Chang[2] and
Hung-Sheng Shang[1]

[1] Division of Clinical Pathology, Department of Pathology, Tri-Service General Hospital, National Defense Medical Center, Taipei, Taiwan, ROC
[2] Division of Infectious Diseases and Tropical Medicine, Department of Medicine, Tri-Service General Hospital, National Defense Medical Center, Taipei, Taiwan, ROC
[3] Division of Pulmonary and Critical Care Medicine, Department of Medicine, Tri-Service General Hospital, National Defense Medical Center, Taipei, Taiwan, ROC
[4] Department of Emergency Medicine, Tri-Service General Hospital, National Defense Medical Center, Taipei, Taiwan, ROC
* These authors contributed equally to this work.

Corresponding authors
Feng-Yee Chang,
fychang@mail.ndmctsgh.edu.tw
Hung-Sheng Shang,
iamkeith001@gmail.com

## ABSTRACT

Coronavirus disease 2019 has become a worldwide pandemic. By April 7, 2020, approximately 1,279,722 confirmed cases were reported worldwide including those in Asia, European Region, African Region and Region of the Americas. Rapid and accurate detection of Severe Acute Respiratory Syndrome Virus 2 (SARS-CoV-2) is critical for patient care and implementing public health measures to control the spread of infection. In this study, we developed and validated a rapid total nucleic acid extraction method based on real-time RT-PCR for reliable, high-throughput identification of SARS-CoV-2 using the BD MAX platform. For clinical validation, 300 throat swab and 100 sputum clinical samples were examined by both the BD MAX platform and in-house real-time RT-PCR methods, which showed 100% concordant results. This BD MAX protocol is fully automated and the turnaround time from sample to results is approximately 2.5 h for 24 samples compared to 4.8 h by in-house real-time RT-PCR. Our developed BD MAX RT-PCR assay can accurately identify SARS-CoV-2 infection and shorten the turnaround time to increase the effectiveness of control and prevention measures for this emerging infectious disease.

## INTRODUCTION

On December 31, 2019, a cluster of pneumonia cases of unknown etiology was reported in Wuhan, Hubei Province, China (*Chan et al., 2020a*). Later, the Chinese Centers for Disease

Control and Prevention (China CDC) reported a novel coronavirus as the causative agent of this outbreak, which was phylogenetically classified into a novel sister clade of SARS virus and named as the severe acute respiratory syndrome coronavirus 2 (SARS-CoV-2). The disease caused by this virus has been called as novel coronavirus disease 2019 (COVID-19).

COVID-19 is an emerging whole world crisis issue. On March 11, 2020, the COVID-19 outbreak was characterized as a pandemic by the World Health Organization. By April 7, 2020, 72,614 fatalities and 1,279,722 laboratory-confirmed cases were reported globally (*Centers for Disease Control & Prevention (CDC), 2020*; *Ghinai et al., 2020*; *World Health Organization (WHO), 2020*; *Wu & McGoogan, 2020*).

A robust, sensitive, specific and high-throughput molecular detection method is urgently needed for SARS-CoV-2 diagnosis. Various methods for detecting SARS-CoV-2 have been reported, including real-time reverse transcription polymerase chain reaction (RT-PCR) and serological testing (*Ai et al., 2020*; *Li et al., 2020b*; *Yu et al., 2020*). Currently, real-time RT-PCR is considered as the gold standard of diagnosis for SARS-CoV-2 because of its high sensitivity; this method can detect SARS-CoV-2 in various samples types, including oropharyngeal swab, sputum and blood (*Wang et al., 2020a*; *Xie et al., 2020*).

The BD MAX System (BD Diagnostic Systems, Franklin Lakes, NJ, USA) is an automated platform which nucleic acid extraction and real-time RT-PCR are performed on the same instrument. This BD MAX System offers not only FDA-cleared panel assays but also an open system mode for user-developed tests (*Hofko et al., 2014*; *McHugh et al., 2018*; *Stokes et al., 2019*; *Widen, Healer & Silbert, 2014*). This study was designed to develop a dual RT-PCR tests for detecting the *E* and *RdRp* genes of SARS-CoV-2 directly from clinical samples using the open system mode of the BD MAX instrument and compare the results to those obtained by the in-house real-time RT-PCR method used in our hospital.

## MATERIALS AND METHODS

### Clinical specimens

This study was approved by Institutional Review Board, Tri-Service General Hospital (TSGHIRB No.: C202005041), registered on 20 March 2020. Informed consent was obtained from patients who signed permission to participate. Throat swab (COPAN COVID-19 Collection & Transport Kits with Universal Transport Medium or Virus Transport Swabs 147C) ($n = 300$) and sputum ($n = 100$) samples were collected from patients with highly suspected travel or contact history in northern Taiwan. Positive control was used with patient's diluted positive RNA aliquoted into Eppendorf tube stored in −80 °C (a range of Ct (threshold cycle) value 34 ± 2 for each run acceptance). Throat swabs were placed in 0.5 mL phosphate-buffered saline and mixed vigorously with a vortex mixer for 30 s to release the cells. Sputum was liquefied by adding an equal volume of 2% *N*-acetyl-ʟ-cysteine in phosphate-buffered saline followed by incubation for 10 min.

**Table 1 Primer and probe sequences used in this study.**

| Target gene | Primer name | Sequence (5′→3′) | References |
|---|---|---|---|
| RdRp (ORF1ab) | RdRp_SARSr-F2 | GTGARATGGTCATGTGTGGCGG | *Corman et al. (2020)* |
| | RdRp_SARSr-R2[†] | CAAATGTTAAAAACACTATTAGCATA | |
| | RdRp_SARSr-P2 | FAM-CAGGTGGAACCTCATCAGGAGATGC-BBQ | |
| *E* | E_Sarbeco_F1 | ACAGGTACGTTAATAGTTAATAGCGT | |
| | E_Sarbeco_R2 | ATATTGCAGCAGTACGCACACA | |
| | E_Sarbeco_P1 | FAM-ACACTAGCCATCCTTACTGCGCTTCG-BBQ | |
| Equine arteritis virus | EAV-IPC-F | CATCTCTTGCTTTGCTCCTT | GenBank EU586274 |
| (EAV) | EAV-IPC-R | AGCCGCACCTTCACATTGAT | |
| IPC | EAV-IPC-P | HEX-CTGACAGCGCTTCTGGTTTCATCAGCT-BHQ | |

**Note:**
[†] Modified the degenerated sequence to fit the Wuhan strain at position 3 from R to A and position 12 from S to A changes marked with an underline.

## In-house SARS-CoV-2 laboratory-developed test (LDT)

### Nucleic acid extraction

Total nucleic acid containing viral RNA was extracted from 0.3 mL of the throat swab supernatant or liquefied sputum, by using LabTurbo Viral nucleic acid extraction kits on a LabTurbo 48 automatic extractor (Taigen Bioscience Corp., Taipei, Taiwan). RNA was eluted with 60 μL of RNase-free water.

### Real-time one-step RT-PCR

The primer and probe sequences of the two target genes (*E* gene as the first-line screening target, followed by confirmatory testing with the *RdRp* gene) has been previously described (*Corman et al., 2020*) (Table 1). The $10\times$ $TCID_{50}$ Equine arteritis virus (EAV) (DIA-EIC; Diagenode, Belgium) was added to extraction buffer for internal control of monitoring RT-PCR inhibitors (Ct value $32 \pm 2$ as acceptance level for each assay). All primers and probes were synthesized and provided by Tib-Molbiol (Berlin, Germany). All assays were performed under the same conditions with some modifications. A 20-μL reaction was prepared containing five μL of RNA, 10 μL of $2\times$ SensiFAST Probe No-ROX One-Step mix (Bioline Reagents Ltd., London, UK), 400 nM of forward and reverse primers, 200 nM of probe, 0.2 μL of reverse transcriptase, and 0.4 μL of RiboSafe RNase inhibitor (Bioline Reagents Ltd., London, UK). Thermal cycling was performed as follows: reverse transcription for 10 min at 50 °C, followed by 95 °C for 2 min and then 50 cycles of 95 °C for 5 s and 58 °C for 30 s on a Rotor-Gene Q real-time PCR machine (Qiagen, Hilden, Germany).

### BD MAX system procedure

We optimized the BD MAX ExK TNA 3 total nucleic acid Extraction Kit on the BD MAX. The BD MAX uses sample buffer tubes containing 750 μL lysis buffer to which the maximal 500 μL primary sample can be added manually. We compared the in-housed SARS-CoV-2 real-time RT-PCR assay with BD MAX System open mode using the same primers, probe and RT-PCR reagents to detect the *E* and *RdRp* genes of SARS-CoV-2. The RT-PCR master mixture without primers & probe (15 μL of $2\times$ SensiFAST Probe

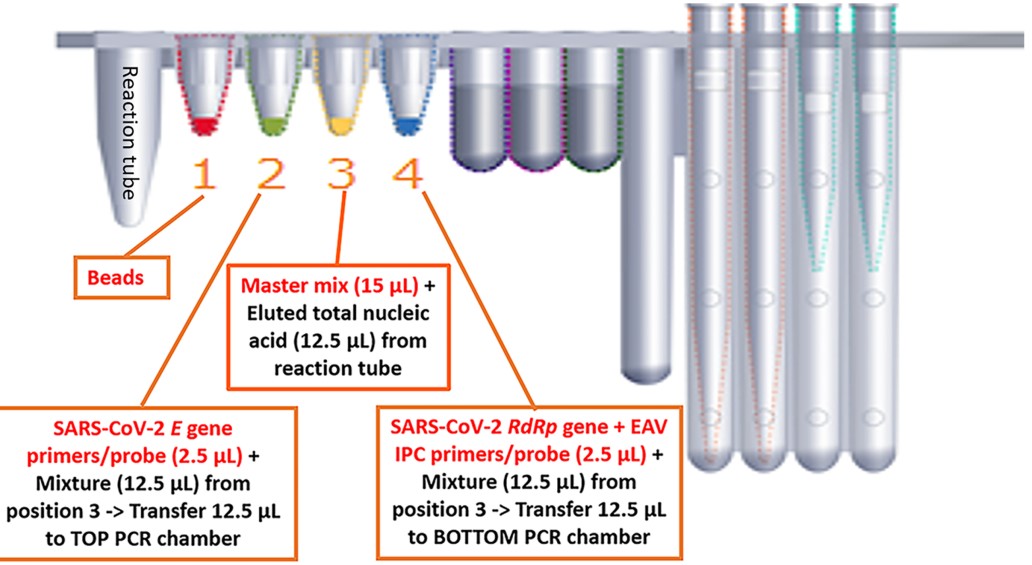

**Figure 1 Experimental design in detecing SARS-CoV-2 on the BD MAX platform.**

No-ROX One-Step mix, 0.3 µL of reverse transcriptase and 0.6 µL of RiboSafe RNase inhibitor) was added to the position 3 on BD cartridge (Fig. 1). The volume of 2.5 µL of primers and probe mixture of *E* gene and *RdRp* + IPC gene was added to the position 2 and position 4. Finally, 12.5 µL eluted nucleic acid was added to position 3 and mix with RT-PCR master mixture. A total of 12.5 µL of above mixture was added to position 2 and mix with 2.5 µL *E* gene primers (2.4 pmole/µL) & probe (1.2 pmole/µL). Residual 12.5 µL mixture was added to position 4 mixing with 2.5 µL *RdRp* + IPC gene primers (2.4 pmole/µL) & probe (1.2 pmole/µL). The mixture in position 2 and 4 were transferred to TOP/BOTTOM PCR chamber (Fig. 1). We carried out the entire sample-to-result procedure in the BD MAX PCR Cartridges. Assay precision was determined by testing individual samples divided into five parts and extracted/assayed separately. The LDT SARS-CoV-2 real-time RT-PCR assay was used as the "gold standard" to assess the diagnostic performance of the BD MAX assay. The BD MAX assay hands-on time was estimated as the sum of the times required to complete sample preparation, device loading and cleaning and result review and reporting.

## RESULTS

### Turnaround time for detecting SARS-COV-2 on the BD MAX System

The procedures used for the BD MAX System included sample preparation, device loading, total nucleic acid extraction, RT-PCR, and results interpretation. The hands-on time and turnaround time between the LDT and BD MAX System were compared (Fig. 2). Use of the BD MAX System improved the turnaround time from 4.8 h to approximately 2.5 h with 24 samples processed simultaneously including prepare samples, clean and prepare rack, create worklist, and prepare MMK (master mix), while also decreasing hands-on time, reducing exposure risk. The BD MAX System can also provide

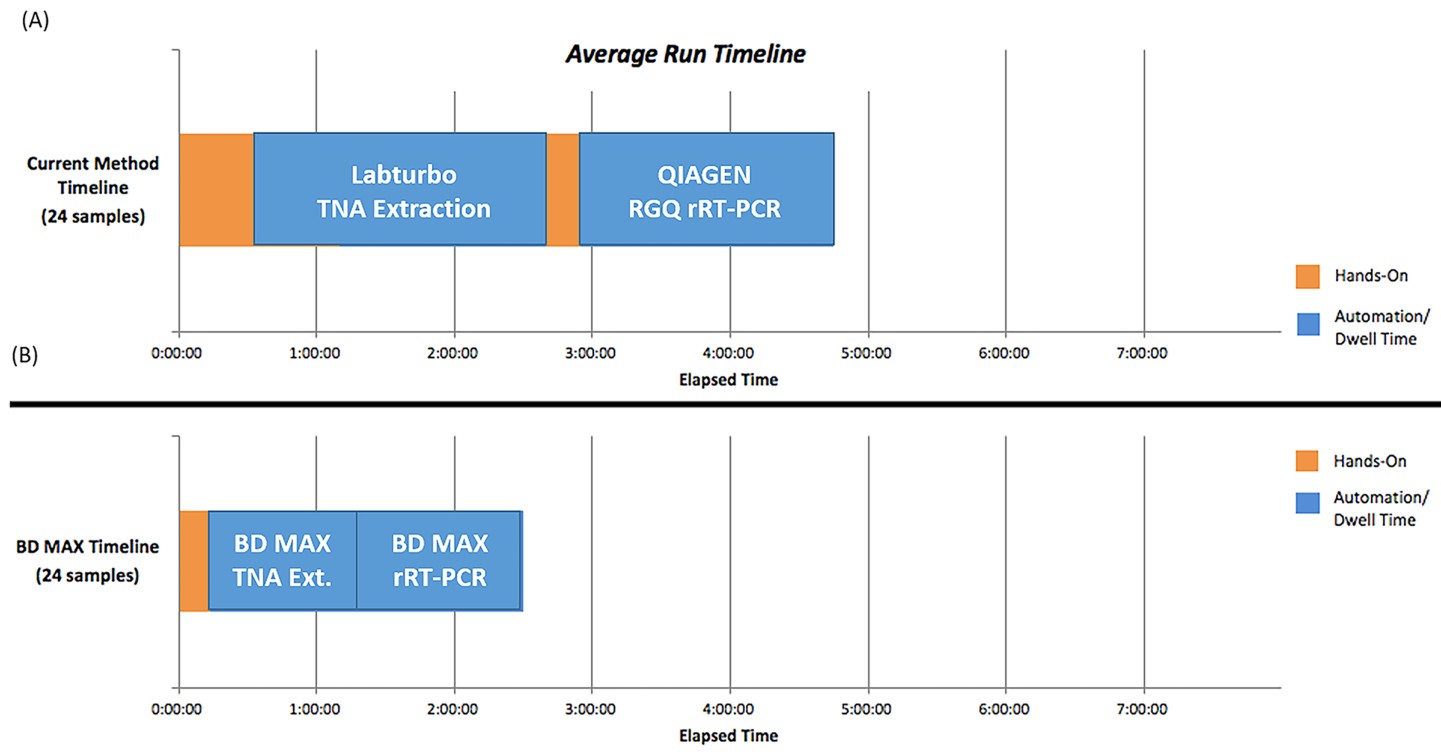

**Figure 2** Hands-on time and TAT (turnaround time) comparison between in-house LDT and BD MAX System (A) In house LDT assay (B) BD MAX assay.

results for 192–216 samples in 11 h (depending on how many batches), showing improved capacity compared to the LDT method.

## Clinical performance of BD MAX System

The analytical specificity of SARS-CoV-2 *E* and *RdRp* gene detection by real-time RT-PCR as well as the cross-reactivity with other respiratory pathogens were determined previously (*Corman et al., 2020*). Here, we focused on the clinical performance of detection using the BD MAX System.

## Reproducibility of BD MAX assay

Five replicates of serial dilutions of SARS-CoV-2-positive samples and a negative template control were tested to evaluate the intra- and inter-assay reproducibility of the BD MAX assay (Table 2). The reproducibility of the Ct values was satisfactory, showing a coefficient of variation of less than 10%.

## Comparison of sensitivity of BD MAX assay and LDT RT-PCR

The empirical sensitivity of the BD MAX assay was determined by evaluating serial dilutions of positive samples and comparing the results to those of the LDT assay. As shown in Fig. 3, diluted specimens were reliably detected by both assays, from the original titer to the $10^{-3}$ titers. BD MAX assay detected four times in eight replicates in

**Table 2 Reproducibility of BD MAX assay for SARS-CoV-2.**

| SARS-CoV-2 | Inter-run | | Intra-run | |
|---|---|---|---|---|
| | No. of positive replicates | Mean Ct ± SD (% coefficient of variation) | No. of positive replicates | Mean Ct ± SD (% coefficient of variation) |
| *E* gene | | | | |
| +++ | 5 | 15.24 ± 0.95 (6.22) | 5 | 16.80 ± 0.95 (5.65) |
| ++ | 5 | 25.06 ± 0.59 (2.35) | 5 | 25.34 ± 0.58 (2.29) |
| + | 5 | 34.68 ± 0.56 (1.62) | 5 | 35.34 ± 0.61 (1.72) |
| *RdRp* gene | | | | |
| +++ | 5 | 17.04 ± 1.12 (6.58) | 5 | 17.26 ± 1.49 (8.65) |
| ++ | 5 | 26.03 ± 0.82 (3.11) | 5 | 26.92 ± 0.74 (2.75) |
| + | 5 | 36.74 ± 0.41 (1.12) | 5 | 37.20 ± 0.43 (1.16) |

**Note:**
+, weak positive; ++, positive; +++, strong positive.

$10^{-4}$ titers which showed less sensitivity compared to LDT assay. Here we used AcroMetrix Coronavirus 2019 (COVID-19) RNA Control (Thermo Fisher Scientific, Waltham, MA, USA) that contain *N*, *S*, *E* and *Orf1ab* regions of SARS-CoV-2 genome for absolute quantification and studying the limit of detection (LOD). Replicate reactions were done at concentrations around the detection end point. The LOD from replicate tests was 8.5 copies per reaction for the *E* gene and *RdRp* gene in LDT assay, while 13.9 copies per reaction for the *E* gene and *RdRp* gene in BD MAX assay.

### Clinical validation of BD MAX assay

A total of 400 clinical samples were included in this study. Most samples were throat swabs ($n = 300$), followed by sputum ($n = 100$). Among the 400 samples, 28 samples were positive and 272 samples were negative for SARS-CoV-2 real-time RT-PCR with the LDT assay and BD MAX System. These 28 positive samples were further confirmed by Taiwan CDC central laboratory. Concordant results were obtained for both assays in SARS-CoV-2 detection, showing 100% agreement (Table 3). The Ct values of the positive specimens for SARS-CoV-2 ($n = 28$) were highly correlated in the comparison of the LDT assay and BD MAX System, with Spearman coefficients of 0.96 and 0.91, respectively (Fig. 4).

## DISCUSSION

This is the first study of the performance of the SARS-CoV-2 detection assay using the BD MAX System. Compared to the in-house results and those from the reference laboratory, the SARS-CoV-2 assay on the BD MAX showed good performance. In this study, we successfully validated a rapid and high-throughput method on the BD MAX platform for accurately and reproducibly identifying SARS-CoV-2 with a greatly reduced turnaround time and fewer hand-on preparation steps including preparing ready-to-use tubes (e.g., 100 Rx without reverse transcriptase stored in −80 °C).

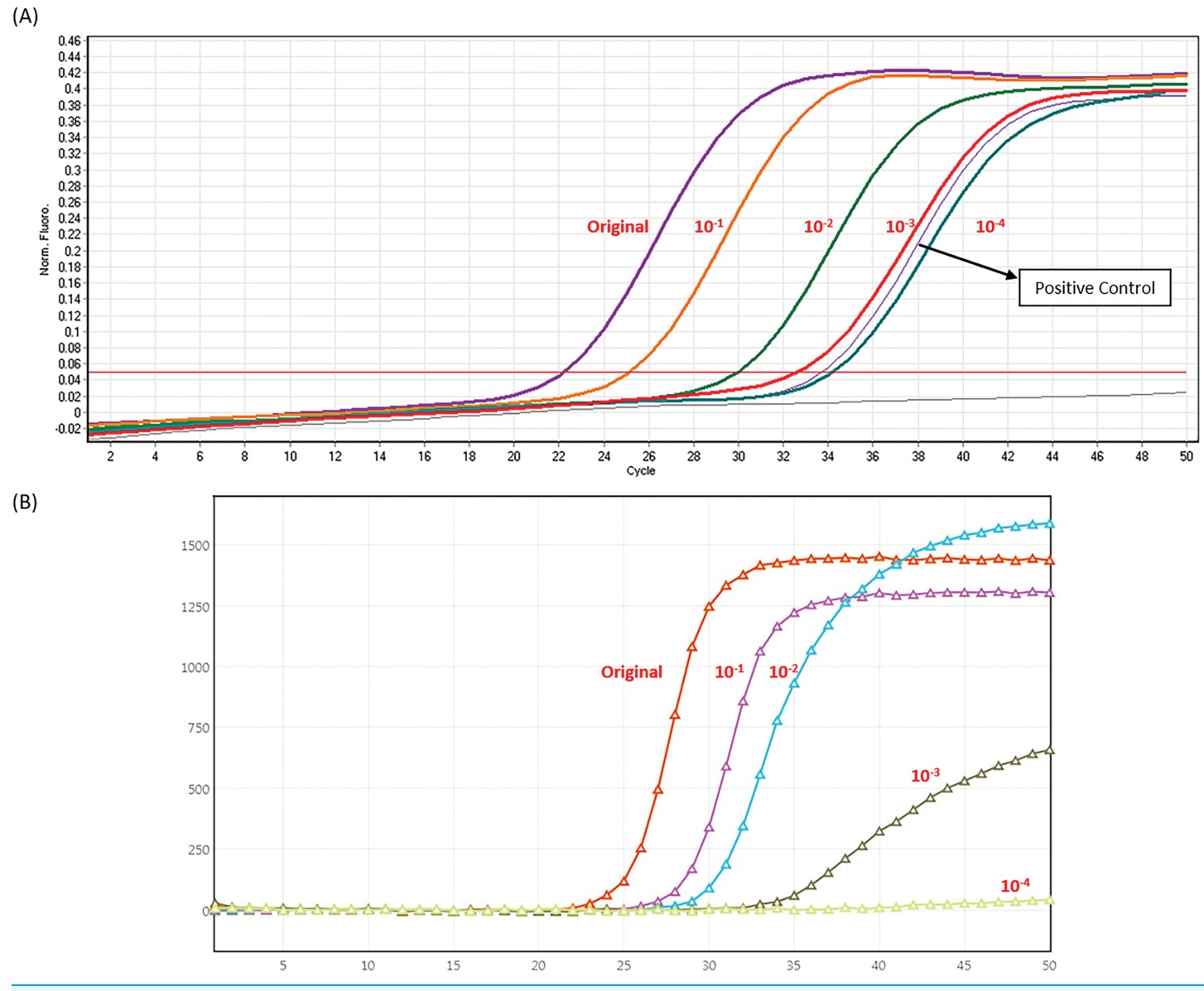

**Figure 3** Analytic sensitivity of LDT and BD MAX System (A) In house LDT assay (B) BD MAX assay.

**Table 3 Comparison of the clinical performance of in-house LDT real-time RT-PCR and BD MAX System for SARS-CoV-2.**

|  |  | BD MAX assay |  |
| --- | --- | --- | --- |
|  | SARS-CoV-2 | Positive | Negative |
| LDT real-time RT-PCR | Positive | 28 | 0 |
|  | Negative | 0 | 372 |

**Note:**
Positive, both *E* gene and *RdRp* gene were detected; Negative, neither *E* gene nor *RdRp* gene were detected.

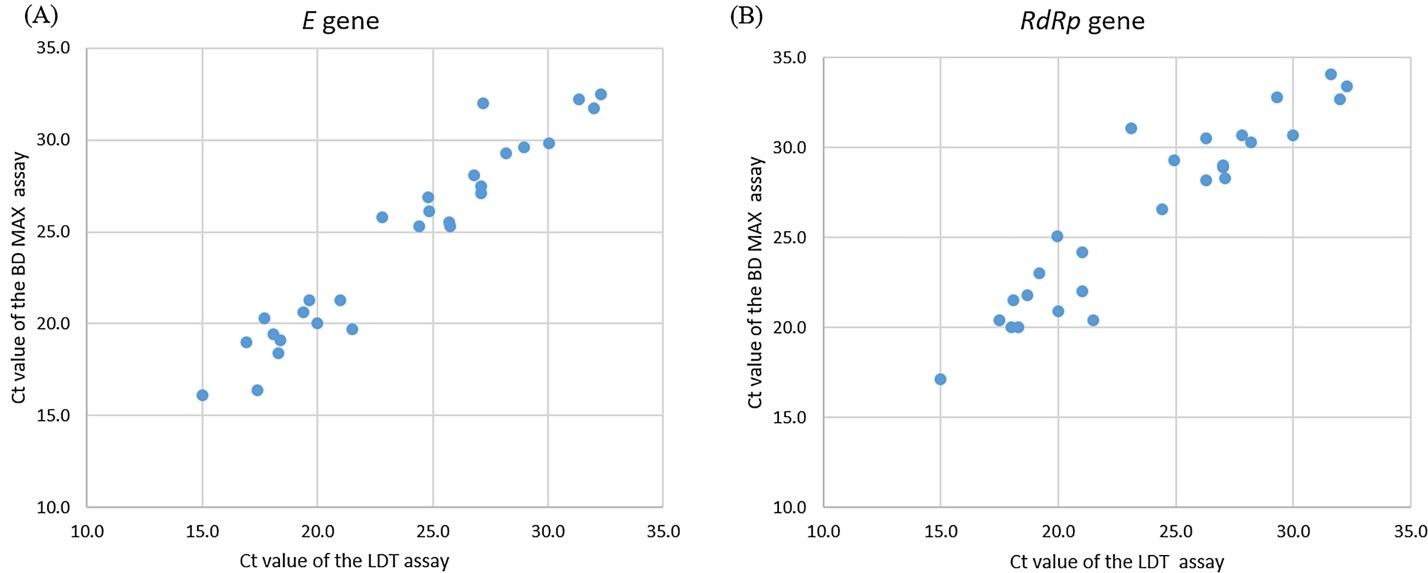

**Figure 4 Correlation of Ct values of clinically positive specimens by LDT and BD Max SARS-CoV-2 assays.** (A) *E* gene for SARS-CoV-2, Spearman coefficient of 0.96. (B) *RdRp* gene for SARS-CoV-2, Spearman coefficient of 0.91.

Several studies have reported that real-time RT-PCR gives false-negative results in detecting SARS-CoV-2 (*Li et al., 2020a*; *Liu et al., 2020*; *Winichakoon et al., 2020*). Combining real-time RT-PCR testing with other clinical features is necessary to diagnose COVID-19 (*Wang et al., 2020b*).

Many detection methods have been used or reported for the diagnosis and/or surveillance of SARS-CoV-2 (*Chan et al., 2020b*; *Corman et al., 2020*; *Wang et al., 2020b*; *Yang et al., 2020*), among which real-time RT-PCR is the most sensitive. Although this method is routinely used, it requires specialized interpretation and has a long turnaround time. The performance of Molecular BD MAX System for different pathogens has been evaluated previously (*Leach et al., 2019*; *Stokes et al., 2019*). These studies demonstrate that the BD MAX System is a good diagnostic tool with a rapid turnaround time, enabling appropriate treatment decisions. During the writing of this manuscript, several molecular assays have been developed for detecting and identifying SARS-CoV-2, including the Cobas SARS-CoV-2 test (Roche, Basel, Switzerland) and TaqPath Covid-19 Combo Kit (Thermo Fisher Scientific, Waltham, MA, USA), which have been authorized for use by the US Food and Drug Administration. To overcome the SARS-CoV-2 pandemic, more sensitive and robust methods are required.

Our designed BD MAX method had several advantages. First, the use of an automated platform with the same primers, probe and master mix as used in our manual LDT assay. Second, there was comparable accuracy, sensitivity and specificity and easy integration into the laboratory workflow. Third, our findings confirm the suitability of the BD MAX system for directly detecting SARS-CoV-2 from clinical specimens. Fourth, the laboratory-developed SARS-CoV-2 BD MAX assay is a dual assay for detecting both *E* and *RdRp* gene, allowing for screening and confirming of SARS-CoV-2 infection

simultaneously. Fifth, the use of only one platform test in routine settings for the clinical diagnosis of emerging infectious agents in various clinical specimens, including sputum and throat swab samples.

## CONCLUSION

In summary, a SARS-CoV-2 real-time PCR molecular test was developed on the BD MAX System. This test showed excellent sensitivity and specificity and can be used to rapidly detect SARS-CoV-2 infection.

Our SARS-CoV-2 test is also rapid and high-throughput, providing accurate and reproducible results with a significantly reduced turnaround time and fewer hands-on preparation steps. This method is very easy with less skillful requirements and can be implemented in emergency medical laboratories for only short training course.

## ACKNOWLEDGEMENTS

We acknowledge and thank technical personnel of BD Diagnostics, for her technical support.

### Funding

This study was supported by Tri-Service General Hospital, Taipei, Taiwan, ROC, Grant Numbers: TSGH-D-109142 and NDMC-NTHU-109-3. The funders had no role in study design, data collection and analysis, decision to publish, or preparation of the manuscript.

### Grant Disclosures

The following grant information was disclosed by the authors:
Tri-Service General Hospital, Taipei, Taiwan, ROC: TSGH-D-109142 and NDMC-NTHU-109-3.

### Competing Interests

The authors declare that they have no competing interests.

### Author Contributions

- Cherng-Lih Perng conceived and designed the experiments, performed the experiments, analyzed the data, prepared figures and/or tables, and approved the final draft.
- Ming-Jr Jian conceived and designed the experiments, performed the experiments, analyzed the data, prepared figures and/or tables, and approved the final draft.
- Chih-Kai Chang conceived and designed the experiments, performed the experiments, analyzed the data, prepared figures and/or tables, and approved the final draft.
- Jung-Chung Lin conceived and designed the experiments, authored or reviewed drafts of the paper, and approved the final draft.
- Kuo-Ming Yeh conceived and designed the experiments, authored or reviewed drafts of the paper, and approved the final draft.
- Chien-Wen Chen conceived and designed the experiments, authored or reviewed drafts of the paper, and approved the final draft.
- Sheng-Kang Chiu conceived and designed the experiments, authored or reviewed drafts of the paper, and approved the final draft.
- Hsing-Yi Chung performed the experiments, prepared figures and/or tables, and approved the final draft.
- Yi-Hui Wang performed the experiments, prepared figures and/or tables, and approved the final draft.
- Shu-Jung Liao performed the experiments, prepared figures and/or tables, and approved the final draft.
- Shih-Yi Li performed the experiments, prepared figures and/or tables, and approved the final draft.
- Shan-Shan Hsieh performed the experiments, prepared figures and/or tables, and approved the final draft.
- Shih-Hung Tsai conceived and designed the experiments, authored or reviewed drafts of the paper, and approved the final draft.
- Feng-Yee Chang conceived and designed the experiments, authored or reviewed drafts of the paper, and approved the final draft.
- Hung-Sheng Shang conceived and designed the experiments, performed the experiments, analyzed the data, prepared figures and/or tables, authored or reviewed drafts of the paper, and approved the final draft.

### Human Ethics

The following information was supplied relating to ethical approvals (i.e., approving body and any reference numbers):

Tri-Service General Hospital Institutional Review Board approved this research (TSGH IRB C202005041).

### Data Availability

Raw data are available in Data S1 and S2.

### Supplemental Information

Supplemental information for this article can be found online at http://dx.doi.org/10.7717/peerj.9318#supplemental-information.

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
