# Peer review of "Novel rapid identification of Severe Acute Respiratory Syndrome Coronavirus 2 (SARS-CoV-2) by real-time RT-PCR using BD Max Open System in Taiwan"

_PeerJ, doi:10.7717/peerj.9318_

## Round 0.1 · original submission · Major Revisions

While the overall reviews for your manuscript were positive, a number of concerns need to be addressed before your submission can be accepted for publication. In particular, increasing the number of positive samples analyzed would significantly strengthen the results.

Reviewer 1 ·

Basic reporting

The report describes the use of the automated BD MAX PCR system for detection of SARS-CoV2 using a lab-developed protocol. Advantages over manual RT-PCR are the decrease in hands-on-times, decrease in turnaround-times and potentially, less contamination risk due to fully automated handling. The authors describe the method and evaluate clinical performance in comparison to the standard lab method employing a validated protocol.

Experimental design

The report is clear and concise, the method and the results are well described. The data set for validation is very limited as only 10 samples among the 300 samples were positive.

Validity of the findings

see below

Additional comments

The following issues should be solved:
• l. 70, what kind of swab have been used to sample the throat?
• l. 74: When using sputa, how many samples were inhibited despite of liquefaction? Did the authors use quality criteria for the internal control (a range of Cts for acceptance)?
• For the turnaround times: Did the authors include the time to prepare the mastermix in the BD MAX hands-on times?
• Remove Figure 3, this information is sufficient as text
• Figure 4: There is a clear reduction in the sensitivity to detect the 1e-4 diluted sample by BD MAX. Also, replicates for this assay comparison are missing. Moreover, for none of both assays there seems to be a good PCR efficiency. The latter can and should be calculated from the dilution series.
• 10 positive samples are not many samples to state 100% agreement as a general statement. The limitation of only 10 pos. samples should be stated in the abstract.
• Discussion: the authors should discuss how to optimize reagent handling to facilitate handling on the BD MAX: Is it possible to prepare read-to-use tubes? Can the mastermix be prepared in advance and stored?
• Figure 5: what are the orange dots? must be removed!

Minor
• l. 36 “and” instead of “band”
• l. 107-112; incomplete sentences, english to be reworked
• BD MAX is not a truly high-throughput machine (change wording)

Reviewer 2 ·

Basic reporting

This manuscript by Perng et al. describes a laboratory-developed test for SARS-CoV-2 on the BD Max system. The manuscript has clear and concise details concerning usage of this laboratory developed test for RNA extraction and RT-PCR analysis, however, key test performance characteristics were not determined, positive sample size (n=10) is too low and data requires reanalysis and reconfiguration prior to resubmission.
Major critiques
- Positive sample size needs to be increased. 10 positive out of 400 samples is insufficient for accurate analysis of clinical sensitivity
o Concordance levels would shift and show variability with lower-level positives from their LDT comparative to the BD Max. The authors (Fig 4) show a clear difference between detection between their assays (BD Max appears less sensitive). The reviewer thinks that the authors should expand this to a larger range of positive samples (10-20 more positives minimum), or use spiked in positive control (plasmid or synthetic RNA) in primary samples to show that at low levels of detection that the concordance remains the same.
- Although precision is presented linearity / limit of detection studies are required to complete analytical test performance characteristics.
- Present in silico or in vitro specificity data with known coronaviruses and other respiratory pathogens to determine analytical specificity
- Internal controls innate to the sample (not spike in external controls) are essential to determine integrity of the RNA in original samples during transport. Consider incorporation of targets such as human Rp
- Define the ORFs themselves used as targets for the PCR – the RNA-dependent RNA-polymerase is a multimeric subunit protein in SARS-CoV2 (nsp7/8 and nsp12) and there have been implications of potentially two subunits with RdRp activity.

Experimental design

Minor critiques
- Clearly define the nature of the positive control used for each assay
- Spell out abbreviations (Ex. LDT (laboratory developed test), TAT (turn-around time)
- Line 136-137 suggest inter and intra-well replicates with data for cross-reactivity analysis. Please show this data.
- Line 26, Severe Acute Respiratory Distress Syndrome Virus 2 should be capitalized as it is the name of the virus.
- Review and correct punctuation and grammar throughout the manuscript

Validity of the findings

see above

Additional comments

see above

Annotated reviews are not available for download in order to protect the identity of reviewers who chose to remain anonymous.

---

## Round 0.2 · accepted · Accept

Thank-you for your submission to PeerJ. I am happy to have been able to work with you towards the acceptance of your manuscript for publication.

Reviewer 1 ·

Basic reporting

My minor initial concerns have been adressed sufficiently. Number of pos. samples has been increased. Although the number is still low, Í think that a well justified first evaluation is now possible. There is a need for SARS-CoV2 assays on automated PCR devices like the BD MAX and thus protocols should be published rapidly.
Would have been fine to also calculate the PCR efficiency, but at least limit of detection is now given.
No further substantial criticism.

Experimental design

ok

Validity of the findings

ok

Reviewer 2 ·

Basic reporting

Addressed all concerns

Experimental design

Addressed all concerns

Validity of the findings

Addressed all concerns

Additional comments

Addressed all concerns